# Biomechanical Evaluation of the Effects of Implant Neck Wall Thickness and Abutment Screw Size: A 3D Nonlinear Finite Element Analysis

**Ming-Dih Jeng [1], Yang-Sung Lin [2] and Chun-Li Lin [2,*]**

1   Department of Dentistry, Far Eastern Memorial Hospital, New Taipei City 220, Taiwan; mdjenq@ms12.hinet.net
2   Department of Biomedical Engineering, National Yang-Ming University, Taipei 112, Taiwan; mergeef@gmail.com
*   Correspondence: cllin2@ym.edu.tw

**Abstract:** In this study, we evaluate the influence of implant neck wall thickness and abutment screw size on alveolar bone and implant component biomechanical responses using nonlinear finite element (FE) analysis. Twelve internal hexagon Morse taper implant–abutment connection FE models with three different implant sizes (diameters 4, 5, and 6 mm), secured with 1.4, 1.6, and 1.8 mm abutment screws to fit with three unilateral implant neck wall thicknesses of 0.45, 0.50, and 1.00 mm, were constructed to perform simulations. Nonlinear contact elements were used to simulate realistic interface fixation within the implant system. A 200 N concentrated force was applied toward the center of a hemispherical load cap and inclined 30° relative to the implant axis as the loading condition. The simulation results indicated that increasing the unilateral implant neck wall thickness from 0.45 to 1.00 mm can significantly decrease implant, abutment, and abutment screw stresses and bone strain, decreased to 58%, 48%, 54%, and 70%, respectively. Variations in abutment screw size only significantly influenced abutment screw stress, and the maximum stress dissipation rates were 10% and 29% when the diameter was increased from 1.4 to 1.6 and 1.8 mm, respectively. We conclude that the unilateral implant neck wall thickness is the major design factor for the implant system and implant neck wall thickness in effectively decreasing implant, abutment, and abutment screw stresses and bone strain.

**Keywords:** biomechanics; dental implant; finite element analysis; implant neck wall; abutment screw

---

## 1. Introduction

Osseointegrated dental implants have been accepted as one of the major treatment concepts for restoring completely and partially edentulous patients [1–6]. The major factor that influences long-term implant use depends mainly on the biological stability provided by the osseointegration between the bone and implant interface (bone/implant interface) [2,5–8]. However, marginal bone remodeling around the implant is affected by implant mechanical behavior even after the bone/implant interface osseointegration is completed [2,5–8]. Overloading leads to marginal bone resorption, usually due to excess mechanical stress transferred from the bone/implant interface onto the supporting bone (crestal cortex) [2,5–9]. Some studies have assessed the implant–abutment connection structure's effects on the marginal bone level change. High stress and marginal bone loss around the external hex connection structure were found [6,10–16]. Besides resorption for the crestal cortex, internal components within the implant (abutment screw) may suffer failure or loosening when unfavorable stress is transferred from the abutment to the implant [2–5,9,17].

Implant–abutment connection design may be a primary factor influencing the force transmission mechanism at the implant–abutment and bone/implant interfaces [2,5,9,17,18]. Different implant–abutment connection designs, i.e., external hex and internal tri/hex connections are predicted to induce different micro motion and stress distribution patterns under occlusal loads [19]. Mechanical complications such as screw fracture or loosening may occur frequently when the occlusal load exceeds the preload applied when the implant–abutment was secured by the retaining screw (abutment screw) [2,5,9,17,18,20,21]. A retrospective study evaluated the implant component fracture type from 1289 implants to suggest directions for successful implant treatment. The study indicated that abutment screw fracture occurred most frequently in the maxillary anterior region [22].

From the biomechanical perspective for internal implant–abutment connection, modifying the implant–abutment connection design, such as by increasing the abutment screw diameter or implant neck wall thickness, may enhance the strength of implant components to reduce the implant failure rate. However, the relationship between the unilateral wall thickness in the abutment, implant neck, and abutment screw diameter is mutually compensated, and few studies have addressed their interactions. The unilateral wall thickness in the abutment and implant neck are reduced when the corresponding hole diameter is increased to fit a larger-sized abutment screw. Mao et al. indicated that the abutment screw diameter should preferably exceed $\Phi$1.4 mm and a unilateral wall thickness of >0.5 mm is an optimal selection for abutments [22]. However, this study only evaluated and compared the influence of abutment and fixation screw sizes for a 4.1 mm diameter implant in a mandibular model. Actually, different implant sizes for real clinical use with respect to different implant neck wall thickness are needed to understand their mechanical interactions.

The aim of this study is to evaluate the mechanical interaction between the abutment screw size and unilateral neck wall thickness of the internal implant–abutment connection for three implant sizes using nonlinear finite element analysis. The main effect of abutment screw size and unilateral implant neck wall thickness on mechanical response (stress and strain) was also computed to identify the importance of each parameter.

## 2. Materials and Methods

### 2.1. Finite Element (FE) Model Generation

A 13 mm long three-piece implant designed with internal hexagon Morse taper abutment connection secured with an abutment screw was selected as the evaluation type in this study. The investigated parameters included three implant sizes (implant diameters 4, 5, and 6 mm; D in Table 1) secured with 1.4, 1.6, and 1.8 mm abutment screw diameters (d in Table 1) to fit with three unilateral implant neck wall thicknesses of 0.45, 0.50, and 1.00 mm (t in Table 1). Twelve combinations of implant diameter, abutment screw size, and unilateral implant neck wall thickness were used (Table 1).

Solid models of the 12 implant systems were constructed using a 3D CAD (computer Aided Design) system (Creo, Parametric Technology Co., MA, USA.) and exported into ANSYS (ANSYS Workbench v14, ANSYS Inc., Canonsburg, PA, USA) to assemble a simplified cylinder supporting bone with 16.5mm length (including a 1.5mm thickness cortical shell) and 10mm diameter. The implant was exposed to the bone level at a length of 3mm, and a hemispherical load cap was designed to cover the abutment according to the ISO14801 standard, which specifies a fatigue test method for single post endosseous dental implants of the transmucosal type and their premanufactured prosthetic components [23] (Figure 1). The test is most useful for comparing endosseous dental implants of different designs or sizes. The load center should also be designed according to the standard located at the free central longitudinal axis intersection of the connecting part and the plane normal to the implant longitudinal axis located 11 mm from the implant support level (Figure 1).

**Table 1.** Combinations of implant sizes (**D**), implant neck wall (**t**), and diameter of abutment size (**d**) and their corresponding finite element (FE) models with node/element numbers used in this study.

| Diameter of Implant (D) (mm) | Thickness of Implant Neck (t) (mm) | Diameter of Abutment Screw (d) (mm) | Node Number | Element Number | Illustration of Implant Designed |
|---|---|---|---|---|---|
| Ø4.00 | 0.45 | Ø 1.40 | 442424 | 296904 | |
| | | Ø 1.60 | 475068 | 319867 | |
| | | Ø 1.80 | 507375 | 342558 | |
| Ø4.00 | 0.50 | Ø 1.40 | 440134 | 294937 | |
| | | Ø 1.60 | 472225 | 317487 | |
| | | Ø 1.80 | 472896 | 317988 | |
| Ø5.00 | 0.50 | Ø 1.40 | 543691 | 369648 | |
| | | Ø 1.60 | 576516 | 392682 | |
| | | Ø 1.80 | 606135 | 413360 | |
| Ø6.00 | 1.00 | Ø 1.40 | 580837 | 395051 | |
| | | Ø 1.60 | 614518 | 418713 | |
| | | Ø 1.80 | 643244 | 438669 | |

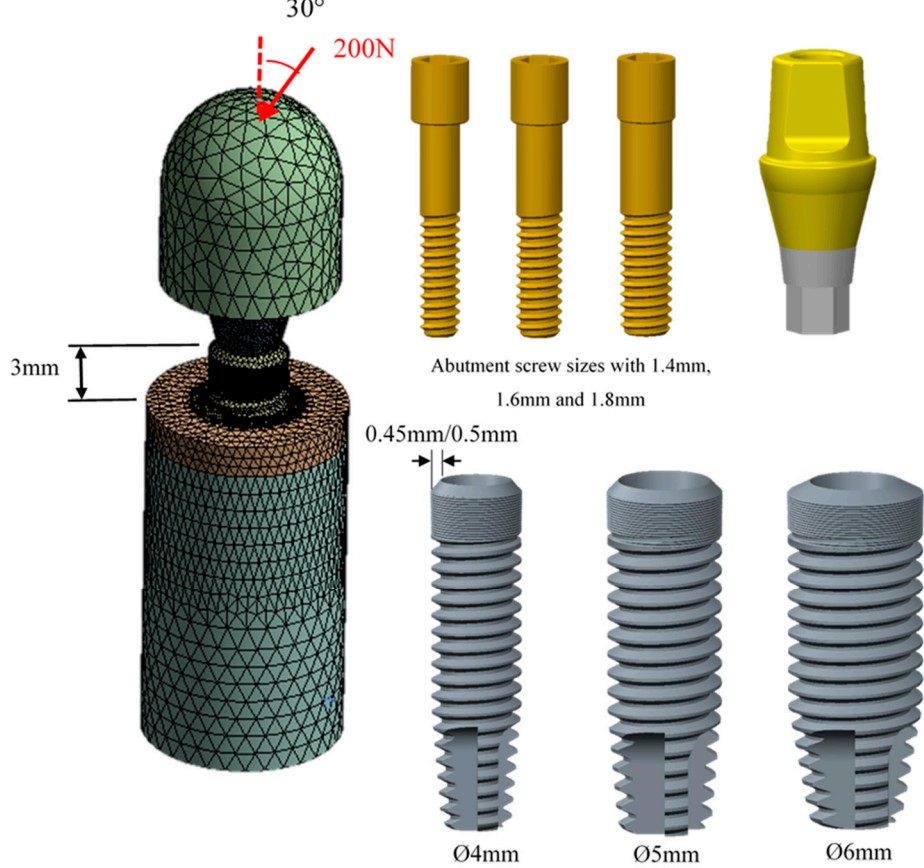

**Figure 1.** Solid models of our simulated models with different implant and abutment screw sizes in the right side and one of the simulated FE mesh models and loading condition in the left side.

Twelve corresponding combination FE mesh models were generated with quadratic 10-node tetrahedral structural solid elements (Solid 92) after the mesh convergence test while controlling for strain energy and displacement variations of <5% for models with different element sizes (Table 1). A bonded condition (continuous displacement) was assumed between the bone and implant to mimic bone/implant interface completion. Nonlinear frictional contact elements (defined as surface-to-surface, i.e., Contact 174/Target 170 in ANSYS) were used to simulate the adaptation between the abutment/abutment screw, abutment screw/implant, and abutment/implant interfaces. The contact

elements allowed the nodes to slip in the tangential direction with no penetration between different materials. Under the frictional condition, the contact zone transfers compressive and tangential forces but no tension [24,25].

## 2.2. Nonlinear FE Simulation

All materials used in the models were considered to be isotropic, homogeneous, and linearly elastic, adopted from the literature (Table 2) [24,25]. The simplified alveolar bone exterior nodes in the FE models were fixed in all directions as the boundary conditions. In order to simulate the preload effect, a 25N-cm torque moment (the manufacturer's recommended tightening torque) was applied to the abutment screw with a friction coefficient of 0.2 assumed for all contact surfaces. This applied moment in the preload stage was transformed along the interface between the abutment screw thread surfaces and implant bore threaded surfaces.

**Table 2.** Material properties adopted in our simulations.

| Material | Young's Modulus (GPa) | Poisson's Ratio |
|---|---|---|
| Cortical bone | 13.7 | 0.3 |
| Cancellous bone | 1.37 | 0.3 |
| Implant (Ti grade 4) | 105 | 0.3 |
| Abutment (Ti6Al4V) | 110 | 0.35 |
| Abutment screw (Ti6Al4V) | 110 | 0.35 |
| Loading device | 193 | 0.26 |

A 200N concentrated force was applied toward the hemispherical load cap center and inclined 30° to the implant axis as the load condition according to the ISO14801 standard. The friction coefficient was set to a value of 0.5 for all contact surfaces [24]. The maximum first principal strain for cortical and cancellous bone was recorded, as was the maximum von Mises stress for the implant, abutment, and abutment screw. The main effect at each level of the two design factors (abutment screw size and implant neck thickness) on the mechanical responses was computed [25]. In order to understand the relative importance of implant neck wall thickness and abutment screw size for all mechanical responses, the data from simulated results were analyzed using a general linear analysis of variance model (ANOVA) test in the MINITAB commercial statistical package (version 12.23, MINITAB Inc., State College, PA, USA) [17].

## 3. Results

Table 3 shows the maximum von Mises stress for the implant, abutment, and abutment screw and the maximum first principal strain for cortical bone. The simulated results show that the unilateral implant neck wall thickness was the main factor which significantly influenced the magnitude of mechanical responses for the implant, abutment, and abutment screw stresses and bone strain (Table 4). Generally, the main effect plot showed that increased unilateral neck wall thickness can decrease the implant, abutment, and abutment screw stresses and bone strain (Figure 2a). When the thickness of the implant neck wall increased from 0.45mm to 1.00mm, the maximum equivalent stresses on the implant, abutment, and abutment screw and the maximum first principal strain of the cortical bone decreased to 58%, 48%, 54%, and 70%, respectively. However, abutment screw size and the interaction between abutment screw size and implant were only significantly influenced by abutment screw stress. The maximum equivalent stress tended to gradually decrease as the screw diameter increased. The decrease rates were 10% and 29% when the screw diameter increased from 1.40 mm to 1.6 mm and 1.80 mm (Figure 2b).

**Table 3.** Simulated results of maximum von Mises stress for the implant, abutment, and abutment screw and maximum first bone strain in all models.

| Diameter of Implant (mm) | Diameter of Abutment Screw (mm) | Neck Thickness of Implant (mm) | Maximum of Equivalent Stress of Abutment Screw (MPa) | Maximum of Equivalent Stress of Abutment (MPa) | Maximum of Equivalent Stress of Implant (MPa) | Maximum of Principle Strain of Cortical Bone (µstrain) |
|---|---|---|---|---|---|---|
| 4 | 1.4 | 0.45 | 876.35 | 726.62 | 536.24 | 8149 |
| 4 | 1.6 | 0.45 | 723.25 | 655.70 | 512.84 | 7053 |
| 4 | 1.8 | 0.45 | 669.79 | 698.81 | 547.70 | 6525 |
| 4 | 1.4 | 0.50 | 725.08 | 820.32 | 556.47 | 7301 |
| 4 | 1.6 | 0.50 | 709.62 | 780.36 | 542.49 | 7048 |
| 4 | 1.8 | 0.50 | 434.73 | 735.41 | 578.34 | 7022 |
| 5 | 1.4 | 0.50 | 633.19 | 359.40 | 327.29 | 3241 |
| 5 | 1.6 | 0.50 | 608.49 | 345.75 | 324.51 | 3327 |
| 5 | 1.8 | 0.50 | 465.29 | 382.16 | 307.73 | 3169 |
| 6 | 1.4 | 1.00 | 415.93 | 351.56 | 214.42 | 2196 |
| 6 | 1.6 | 1.00 | 335.55 | 361.90 | 224.69 | 2155 |
| 6 | 1.8 | 1.00 | 301.44 | 373.09 | 236.83 | 2075 |

**Table 4.** The ANOVA test *p*-values ($p < 0.05$ shows significance).

|  | Implant Stress | Abutment Stress | Abutment Screw Stress | Bone Strain |
|---|---|---|---|---|
| Implant neck wall thickness | <0.001 | 0.039 | 0.002 | 0.002 |
| Abutment screw size | <0.001 | 0.961 | 0.960 | 0.817 |
| Implant neck wall thickness * Abutment screw size | 0.021 | 0.999 | 0.998 | 0.977 |

The highest von Mises stresses were found on the compressive side at the neck region for implant thread region tension side for the abutment screw and the hexagon connection junction for the abutment. Also, the highest bone strain values were found at the crestal cortex region that received tensile stress (Figure 3).

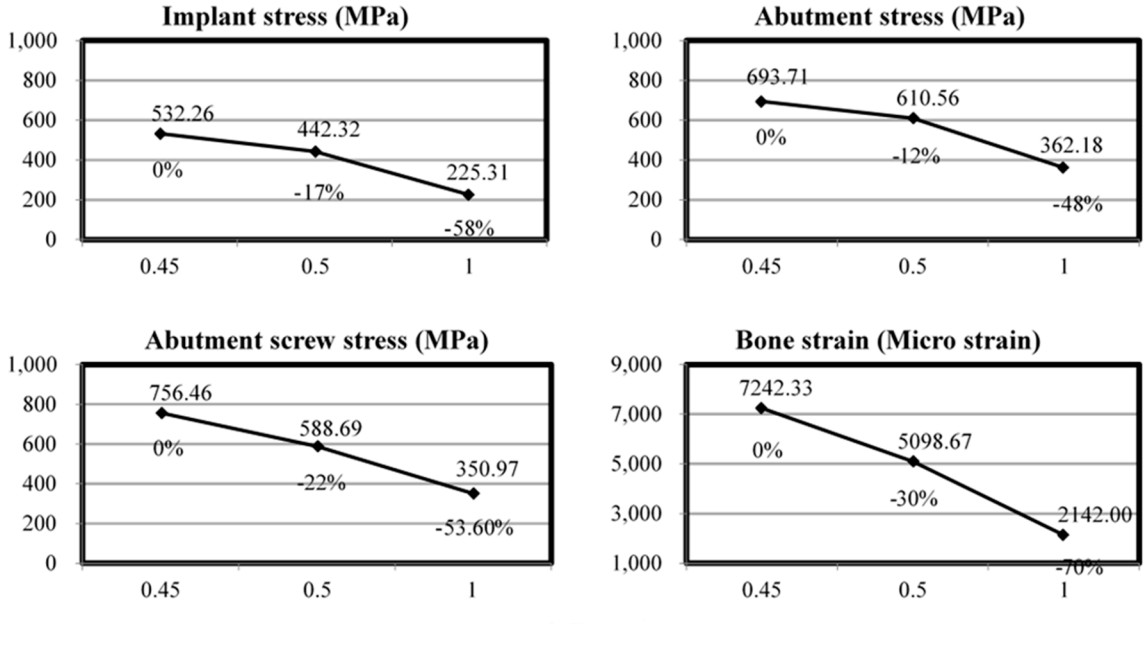

(a)

**Figure 2.** *Cont*.

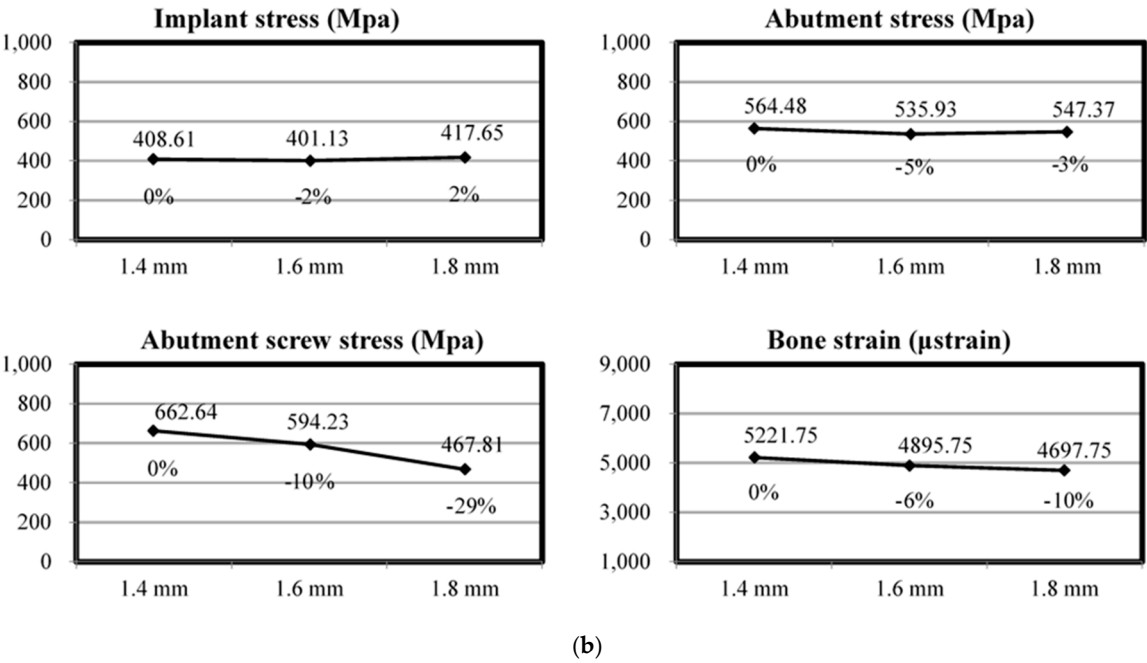

**Figure 2.** Main effect plots of implant neck wall (**a**) and abutment screw size at each level for maximum implant, abutment, and abutment screw stresses and maximum bone strain (**b**).

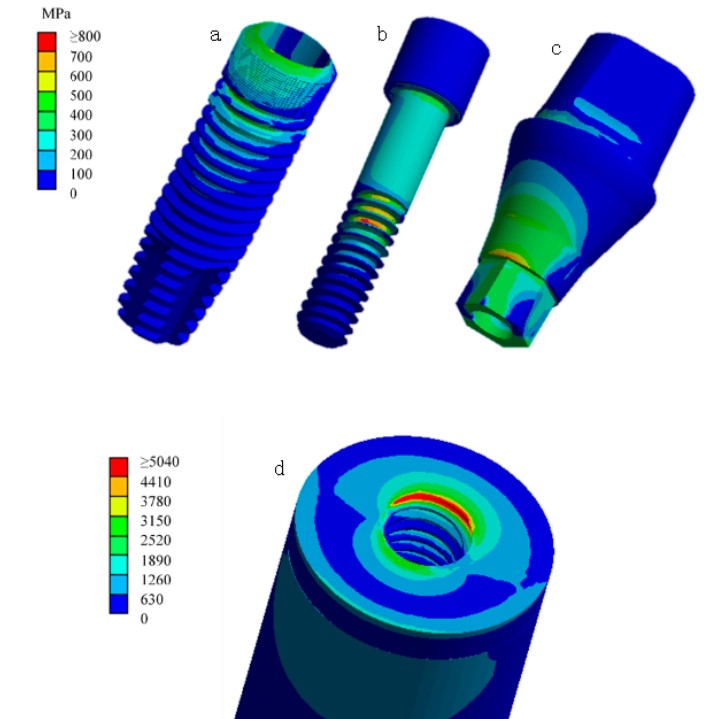

**Figure 3.** Stress patterns for implant stress (MPa) (**a**), abutment screw stress (MPa)(**b**), abutment stress (MPa) (**c**), and bone strain (micro strain) (**d**).

## 4. Discussion

The goal of dental implant development is to find those conditions that have better optimal structure for favorable biomechanical transmission. The stress of each component within the implant system cannot come close to or be greater than the ultimate implant material strength. The stress transmitted from the implant to the bone must also be as favorable as possible because excess load may

easily result in poor osseointegration, failing the major design criterion [2,5–8]. However, complicated biomechanical complications are often caused by multiple factors and their interactions, and it is difficult to evaluate and predict them solely by clinical study or experiments [9].

Experimental approaches or clinical observations cannot provide enough information to determine the detailed mechanical response and interactions relating to variations in the structure design parameters, such as the abutment screw size and implant neck thickness investigated in this study. Nonlinear FE analysis with reasonable interface conditions (contact) that could simulate the inherent flexibility within the implant system was employed to understand the biomechanical influence of abutment screw size with a hexagonal Morse taper three-piece internal connection implant and implant neck thickness on the surrounding bone. The finite element (FE) method can alter the parameters in a more controllable manner and has become a commonly used analytical tool in orthopedic/dental biomechanical studies [24–26]. Nevertheless, most related research focused only on different implant–abutment connections, i.e., internal friction and external hex connections, to influence the peri-implant crestal bone resorption [2,5–8]. However, many clinical trials have pointed out that abutment and abutment screw damage are the other two main reasons for implant failure [2,5–8].

The objective of this study was to evaluate and compare the mechanical effects of the implant neck wall thickness and abutment screw diameter on stability using 3D CAD technology and nonlinear FE analysis. We referred to the ISO14801 standard force application method and applied oblique force at a 30° angle to the implant axis to simulate the situation where a single tooth is severely oblique in a clinical situation. A load of 200 N is considered to be about two-thirds of the maximum occlusal force in the posterior teeth [23]. Our FE model also considered cortical and cancellous bones to mimic real anatomical status for elevating clinical responses.

The unilateral implant neck wall thickness significantly influenced the mechanical response magnitude for the abutment (12%), abutment screw (22%), and implant (17%) stresses and bone strain (30%), which were observed to decrease by at least 12% when the wall thickness was increased from 0.45 mm to 0.5 mm (Figure 2a). This phenomenon was due to the increased structure stiffness at the implant neck, able to disperse the stress transmission from the abutment to the implant and from the abutment to the screw. The stress concentration was also reduced at the hexagon connection junction because a thickened implant neck wall can distribute more stress to the abutment Morse taper. The decrease in bone strain might also be due to a stronger implant neck wall causing difficulty in the force pushed and squeezed from implant to bone.

ANOVA analysis showed that the abutment screw size influenced the structure stiffness, and the maximum stress decrease rates were 10% and 29% when the diameter increased from 1.4 mm to 1.6 mm and 1.8 mm, respectively. The percentage variations in implant and bone strain values of maximum stress were found to be smaller than 10% (Figure 2b). This might be due to discontinued stress transfer from the abutment screw to the implant/surrounding bone. Abutment stress only slightly (≤3%) decreased when the screw diameter was increased because the stiffness in the whole abutment structure was much higher than that for the abutment screw.

Some limitations still exist in this numerical study. The load condition simulated in the FE analysis was not realistic and approximated only lateral occlusal force, which is more dangerous clinically. The simplified crown and bone model geometry was simulated based on Saint Venant's principle, and the whole mandibular body is very elaborate [26]. Implants smaller than 4mm in diameter, usually used in anterior teeth, might not suitably fit the result found in our simulations because the designed space for the implant neck thickness and the screw diameter were limited. All material properties were elastic and assumed to be homogeneous and isotropic. The mechanical responses obtained from all simulations were a first approximation and must be validated with clinical trials.

## 5. Conclusions

The unilateral implant neck wall thickness was found to be the major design factor for the implant system. We conclude that thickening the implant neck wall as much as possible can more effectively decrease implant, abutment, and abutment screw stresses and bone strain.

**Author Contributions:** Conceptualization of the research, M.-D.J. and C.-L.L.; methodology, Y.-S.L. and C.-L.L.; software, Y.-S.L.; experimental data analysis, Y.-S.L. and C.-L.L.; writing—original draft preparation, C.-L.L.; writing—review and editing, M.-D.J. and C.-L.L. All authors have read and agreed to the published version of the manuscript.

**Funding:** This study was supported in part by MOST108-2622-E-010 -001-CC2, Ministry of Science and Technology, Taiwan and by the Far Eastern Memorial Hospital and National Yang-Ming University cooperation research project, Taipei, Taiwan.

**Conflicts of Interest:** The authors declare no conflict of interest.

**Data Availability Statement:** The experimental data (Tables 1 and 3) used to support the findings of this study are included within the article. We agree to share our experimental data for research to verify our results, replicate the analysis, and conduct secondary analyses. Please contact: Chun-Li Lin, Ph.D., Professor.

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
