# Peer review of "Biomechanical Evaluation of the Effects of Implant Neck Wall Thickness and Abutment Screw Size: A 3D Nonlinear Finite Element Analysis"

_applsci, doi:10.3390/app10103471_

Round 1

Reviewer 1 Report

The article suits the general topic of the journal and shows a suitable observational study in the field of biomechanical analysis of implants, focusing on implant wall thickness and abutment screw size.

The language, structure and design are appealing. The background research performed was good and the information provided are sufficient.

Even though there are a minor and a major concern.

The major concern in my opinion relates to the generation of the FE model. As described in Table 1, the numbers of node and element are decreasing in the 4,0mm implant when the neck thickness is increasing. I can't understand this relation, when the total surface is increasing. I would recommend to check the data again and give an explanation for this. 

The minor concern relates to the attached figure 1.

I would recommend to improve the labeling, by placing the exact numbers, next to each screw or implant.

Thank you very much and I am happy to review your paper again.

Author Response

Reviewer #1:

General Comments

The article suits the general topic of the journal and shows a suitable observational study in the field of biomechanical analysis of implants, focusing on implant wall thickness and abutment screw size. The language, structure and design are appealing. The background research performed was good and the information provided are sufficient.

Response:

Thanks to the reviewer’s comment and encouragement.

Specific comments:

Even though there are a minor and a major concern.

Point 1:

The major concern in my opinion relates to the generation of the FE model. As described in Table 1, the numbers of node and element are decreasing in the 4,0mm implant when the neck thickness is increasing. I can't understand this relation, when the total surface is increasing. I would recommend to check the data again and give an explanation for this.

Response 1:

The numbers of node and element in finite element model is not necessarily related to surface area, but is related to geometry features, such as hole/groove/fillet/sharp point that change geometry feature greatly or are prone to stress concentration, the mesh number should be increased. The “Smart mesh” function provided in ANSYS was used to perform mesh procedure for our models, this function can judge and control mesh quality according to the abfore mentioned condition. Otherwise, numbers of node and element were correct because smaller abutment was needed when the neck wall thickness increased in the 4,0mm implant.  One Figure of implant designed illustration included in original Table 1 was loss by Editor office and might induce the Reviewers to confusion and has been corrected in the revised manuscript.

Point 2:

The minor concern relates to the attached figure 1.

I would recommend to improve the labeling, by placing the exact numbers, next to each screw or implant.

Response 2:

The figure caption of Figure 1 was corrected according to the Reviewer’s suggestion.

Reviewer 2 Report

the work is not original in the theme, widely discussed in literature, however the analysis system is very precise and appropriate. through careful analysis, in fact, very interesting conclusions are reached for the reader. the most important thing is the final concept that the thickening of the implant neck wall as large as possible can more effective decrease implant, abutment, abutment screw stresses and bone stain.

an interesting work

Author Response

General Comments

The work is not original in the theme, widely discussed in literature, however the analysis system is very precise and appropriate. through careful analysis, in fact, very interesting conclusions are reached for the reader. the most important thing is the final concept that the thickening of the implant neck wall as large as possible can more effective decrease implant, abutment, abutment screw stresses and bone stain.

an interesting work

Response:

Thanks to the reviewer’s comment, we agree this work is not original in the theme. However, this study suggested that the unilateral implant neck wall thickness was the major design factor for the implant system and should be designed to increase its thickness as much as possible to reduce failure rate.